# Similar effects of edaphic and climatic factors on soil organic carbon stocks of the world

Zhongkui Luo[1], R. A. Viscarra Rossel[2], Tian Qian[1]

[1]College of Environmental and Resource Sciences, Zhejiang University, Hangzhou, Zhejiang 310058, China

[2]Soil and Landscape Science, School of Molecular & Life Sciences, Curtin University, Perth, WA 6845, Australia

*Correspondence to:* Zhongkui Luo (luozk@zju.edu.cn)

**Abstract.** Soil organic carbon (SOC) accounts for two-thirds of terrestrial carbon. Yet, the role of soil physiochemical properties in regulating SOC stocks is unclear, inhibiting reliable SOC predictions under land use and climatic changes. Using legacy observations from 141,584 soil profiles worldwide, we disentangle the effects of biotic, climatic and edaphic factors (a total of 31 variables) on the global spatial distribution of SOC stocks in four sequential soil layers down to 2 m. The results indicate that the 31 variables can explain 60-70% of the global variance of SOC in the four layers, to which climatic variables and edaphic properties each contribute ~35% except in the top 20 cm soil. In the top 0-20 cm soil, climate contributes much more than soil properties (43% vs 31%); while climate and soil properties show the similar importance in the 20-50, 50-100 and 100-200 cm soil layers. However, the most important individual controls are consistently soil-related, and include soil texture, hydraulic properties (e.g., field capacity) and pH. Overall, soil properties and climate are the two dominant controls. Apparent carbon inputs represented by net primary production, biome type and agricultural cultivation are secondary and their relative contributions were ~10% in all soil depths. This dominant effect of individual soil properties challenges current climate-driven framework of SOC dynamics, and need to be considered to reliably project SOC changes for effective carbon management and climate change mitigation.

## 1 Introduction

Soil organic carbon (SOC) represents the largest pool of terrestrial carbon (Le Quéré et al., 2016;Batjes, 2016) and plays a key role in combating climate change and ensuring soil productivity. To better manage land for maintaining SOC levels or enhancing carbon sequestration, it is vital to elucidate controlling factors of SOC stabilization and stock. As an important soil property, it is reasonable to expect that SOC might be integrally influenced by five predominant factors controlling soil development and formation; namely, climate, organisms, topography, parent materials, and time (Jenny, 1941). However, climate is usually prioritized and considered to be critical (Carvalhais et al., 2014) because of its direct effect on soil carbon inputs via photosynthetic carbon assimilation, and output via microbial decomposition. But climate-driven predictions of SOC dynamics (e.g., using Earth system models) remain largely uncertain, particularly across large extents (Todd-Brown et al., 2013;Bradford et al., 2016;Luo et al., 2017).

A primary source of the uncertainty is our poor understanding of how edaphic properties regulate SOC stabilization and stock in soil (Davidson and Janssens, 2006;Dungait et al., 2012). For example, SOC can be physically protected from decomposition via occlusion within soil aggregates and adsorption onto minerals (Six et al., 2000), which create physical barriers preventing microorganisms to decompose carbon sources (Doetterl et al., 2015;Schimel and Schaeffer, 2012) regardless of climate conditions, but how this protection influences global SOC stocks is unclear. Additionally, the soil physicochemical environment controls the supply of water, nutrients, oxygen and other resources, which are required for microbial communities to utilize SOC as well as for plant carbon assimilation to replenish soil carbon pool. Considering the large spatial variability of soil properties globally, we need to understand the edaphic controls of SOC better. By explicitly considering the effect of soil physicochemical properties, we hope to promote a review of climate-driven frameworks of SOC dynamics.

In addition to our incomplete understanding of the general importance of soil properties in regulating SOC stocks, whether and how their effects vary with soil depth are also unclear. Most studies focus on topsoil layers (e.g., 0–30 cm), even though globally, deeper soil layers (below 30 cm) store more carbon than topsoils (Jobbágy and Jackson, 2000;Batjes, 2016). Like the topsoil SOC pool, the subsoil SOC pool may actively respond to climate and land use changes. Studies of whole soil profiles have observed increased loss of subsoil SOC under warming (Pries et al., 2017;Melillo et al., 2017;Zhou et al., 2018) as well as under additional supply of fresh carbon (Fontaine et al., 2007). Land uses such as cropping and grazing can also induce substantial subsoil SOC loss (Sanderman et al., 2017), which is concerning because of the potential adverse effects of climate and land use changes. It is therefore imperative that we better understand the controlling factors of SOC in deep soil layers as this will help to develop unbiased strategies to effectively manage whole-soil profile carbon.

Here, we aim to disentangle the relative importance of climatic, biotic and edaphic controls on SOC stocks globally in different soil layers. To do so, we assessed data from 141,584 whole-soil profiles across the globe including measurements of SOC and other soil physicochemical properties, collated by the World Soil Information Service (WoSIS) (Batjes et al., 2017). For each profile, 19 climate-related covariates reflecting seasonality, intra- and inter-annual variability of climate were obtained from the WorldClim database (Fick and Hijmans, 2017), the MODIS NPP (net primary productivity) product (Zhao and Running, 2010) was used to infer apparent carbon input into soil, and the MODIS land cover product (Channan et al., 2014) to obtain land use information. Using these data sets, we disentangled the relative importance of biotic, climatic and edaphic covariates (a total of 31 variables, Table 1) in controlling the spatial variance in SOC stocks worldwide in four sequential soil layers (i.e., 0–20, 20–50, 50–100, and 100–200 cm), and identified the correlations between SOC stock and the most important variables.

## 2 Materials and Methods

### 2.1 Observed soil profile data and harmonization

The World Soil Information Service (WoSIS) collates and manages the largest database of explicit soil profile observations across the globe (Batjes et al., 2017) which forms the foundation of a series of digital soil mapping products such as the global SoilGrids (Hengl et al., 2017). The WoSIS dataset is still growing. When we visited the dataset last on 25 March 2019, there were a total of 141,584 profiles which were used in this study. These profile observations were quality-assessed and standardized, using consistent procedures (Batjes et al., 2017). In each soil profile, multiple layers were sampled for determining SOC content and/or other soil properties. A total of 48 soil properties were recorded with multiple variates of the same property (e.g., pH measured in $H_2O$, $CaCl_2$, KCl etc.). In the data assessment, we considered nine principal soil physicochemical properties other than SOC itself in the data analysis (Table 1). Taking the advantage of all measurements, however, other soil properties were used for missing data imputation (see the section 2.2). The layer depths are inconsistent between soil profiles. We harmonized all soil properties including SOC to four standard depths (i.e., 0–20 cm, 20–50 cm, 50–100 cm, and 100–200 cm) using mass-preserving splines (Bishop et al., 1999;Malone et al., 2009). This harmonization enables the calculation of SOC stock in the defined standard layers, making it possible to directly compare among soil profiles.

## 2.2 SOC stock calculation and filling missing values

We calculated SOC stock ($SOC_s$, kg C m$^{-2}$) in each standard depth as:

$$SOC_s = \frac{OC}{100} \cdot D \cdot BD \cdot \left(1 - \frac{G}{100}\right), \tag{1}$$

where $OC$ is the weight percentage SOC content in the fine earth fraction <2 mm, $D$ the soil depth (i.e., 0.2, 0.3, 0.5 or 1 m in this study), $BD$ the bulk density of the fine earth fraction <2 mm (kg m$^{-3}$), and $G$ the volume percentage gravel content (> 2 mm) of soil. Amongst the 141,584 soil profiles, unfortunately, only 9,672 profiles have all the measurements of $OC$, $D$, $BD$ and $G$ to enable direct calculation of SOC stock. We call these profiles "stock profiles".

Another 82,734 profiles have measured $OC$ (i.e., the weight percentage SOC content), but $BD$ and/or $G$ are missing. We call these profiles "content profiles". To utilize and take advantage of all $OC$ measurements, we used generalized boosted regression modelling (GBM) to perform imputations (i.e., fill missing data). As such, $SOC_s$ can be estimated. To do so, for $BD$ and $G$ in each standard soil depth, GBM was developed based on all measurements of that property (e.g., BD) in the 141,584 profiles with other 32 soil properties. Total carbon which includes organic and inorganic carbon, and another nine soil properties (Table 1) which were used as predictors of SOC stocks were excluded as covariates (i.e., predictors). The final GBM model was validated using 10-fold cross-validation repeated 10 times, and applied to predict missing values of BD and G. A total of 92,406 "SOC profiles" including "stock profiles" and "content profile" with relevant measurements of other nine soil properties (Table 1) was obtained, and used to assess the effects of various variables on $SOC_s$ (section 2.4). These "soil profiles" cover 13 major biome groups although the profile numbers vary from 472 in flooded grasslands and savannas to 24,382 in temperate broadleaf and mixed forests (Fig. 1). The profiles also cover various climate conditions across the globe with the mean annual temperature ranging from –19.6 to 30.7 °C and mean precipitation ranging from 0 to 667.4 cm yr$^{-1}$ (Fig. 1). The

prediction error of the GBM were propagated into the calculation of $SOC_s$ to account for uncertainty resulting from data imputation (see section 2.4).

## 2.3 Biotic and climatic covariates

For each "SOC profile", NPP was extracted from MODIS NPP product (Zhao and Running, 2010). The NPP product includes the annual NPP from 2001 to 2015 at the resolution of 1 km$^2$, which was estimated by analysing satellite data from MODIS using the global MODIS NPP algorithm (Zhao et al., 2005;Zhao and Running, 2010). NPP is the net carbon gained by plants (i.e., the difference between gross primary productivity and autotrophic respiration). If assuming a steady state of the vegetation (i.e., no long-term directional change of carbon biomass in plants), NPP will end up in soil via rhizodeposition and litter fall,

and equals to total carbon input into soil. Here we calculated the average NPP based on the data from 2001 to 2015, and called this average NPP the apparent carbon input to soil, acknowledging that not all ecosystems are at strict steady state, particularly those ecosystems (e.g., croplands) actively interact with human activities. The WWF (World Wildlife Fund) map of the terrestrial ecoregions of the world (Olson et al., 2001; https://www.worldwildlife.org/publications/terrestrial-ecoregions-of-the-world) was used to extract the biome type for each location. The MODIS land cover map (Channan et al., 2014) at the

same resolution of NPP databases was used to identify that if the land is cultivated (i.e., land cover type of croplands and cropland/natural vegetation mosaic) at the location of each soil profile.

In addition to NPP, land cover and biome type, 19 climatic variables (Table 1) for each "SOC profile" were obtained from the WorldClim version 2 (Fick and Hijmans, 2017). The WorldClim version 2 calculates biologically meaningful variables using monthly temperature and precipitation during the period 1970-2000. The data at the same spatial resolution of

the NPP data (i.e., ~1 km$^2$) was used in this study. Eleven of the19 climatic variables are temperature-related (Table 1), and eight are precipitation-related (Table 1). These variables reflect the seasonality, intra- and inter-annual variability of climate, which would have both direct (via decomposition thus carbon outputs from soil) and indirect (via carbon assimilation thus carbon inputs to soil) effect on SOC stock.

## 2.4 Data analysis

A machine learning-based statistical model - boosted regression trees (BRT) – was performed to explain the variability of $SOC_s$ across the globe and identify important controlling factors. A big advantage of the BRT model is its ability to model high-dimensional data set, taking into account nonlinearities and interplay (Elith et al., 2008). Using the BRT model, we modelled $SOC_s$ in each standard depth as a function of edaphic variables in that depth, climatic and biotic variables (Table 1):

$$SOC_s = f(edaphic, climatic, biome, NPP, cultivation). \tag{2}$$

We used a 10-fold cross-validation to constrain the BRT model in R 3.6.1 (R Core Team 2019) using algorithms implemented in the R package *dismo*. The amount of variance in SOC$_s$ explained by the model was assessed by the coefficient of

determination ($R^2$). To assess the potential uncertainty induced by the uneven distribution of soil profiles across the globe as well as the imputation of missing BD and G for estimating $SOC_s$, we conducted 200 bootstrapping simulations (i.e., resample all soil profiles with replacement). For each bootstrap sample, $SOC_s$, if BD and G are missing, was recalculated using BD and G imputed by GBM plus an error randomly sampled from the distribution of imputation error. Using the new $SOC_s$ estimations, then, a new BRT model was fitted.

Considering the potential collinearity in the 19 climatic variables as well as in the nine soil properties, the BRT model was conducted using their principle components. That is, a principle component analysis (PCA) was performed to eliminate potential correlations in the soil and climatic variables, respectively. The important principal components (PCs) with variances of greater than 1 were retained in the BRT model based on Kaiser's criterion (Kaiser 1960). The PCA was performed using the function prcomp in the package stats in R 3.6.1 (R Core Team, 2019). In addition, in order to demonstrate the importance of soil properties, we fitted another set of BRT model without soil properties. The model performance with and without soil properties were compared in terms of explaining the variance of SOC stocks across the globe.

The BRT model allows the estimation of the relative influence of each individual variable in predicting $SOC_s$, i.e., the percentage contribution of variables in the model. The relative influence is calculated based on the times a variable selected for splitting when growing a tree, weighted by squared model improvement due to that splitting, and then averaged over all fitted trees which were determined by the algorithm when adding more trees cannot reduce prediction residuals (Elith et al., 2008;Friedman and Meulman, 2003). As such, the larger the relative influence of a variable, the stronger the effect on $SOC_s$. In addition, we also calculated the 95% confidence intervals as the 2.5% and 97.5 quantiles of the relative influence estimated by 200 bootstrapping simulations, which represent the uncertainty in the importance of variables. To facilitate interpretation, the relative influence of each variable is scaled so that the sum of the influence of all variables is equal to 100. The overall relative influences of edaphic (i.e., the sum relative importance of all soil-related variables) and climatic (i.e., the sum relative importance of all climate-related variables) variables as well as biome type, NPP and cultivation were also calculated and compared. As we have 200 estimations (i.e., 200 bootstraps) of the relative influence, we calculated a weighted average relative influence for each variable with weights based on the $R^2$ of each BRT model.

## 3 Results

The 19 climatic variables could be represented by four principle components (PCs, i.e., Climate1-4 which were selected by Kaiser's criterion) which could explain 88% of their variance (Fig. 2; only were the first two PCs shown); and 72% of the variance in nine soil properties could be explained by three PCs (i.e., Soil1-3, Fig. 2). For Climate1-4, the most important contributing variables were T11 (mean temperature of coldest quarter), P6 (precipitation of driest quarter), T5 (max temperature of warmest month) and P7 (precipitation of warmest quarter), respectively. For Soil1-3, the most important contributing variables were sand content, pH and silt content, respectively (Fig. 2). Using Climate1-4, NPP, biome type and cultivation as predictors, the BRT model could explain 53%, 46%, 42%, and 49% of the variance of SOC stocks in the 0-20,

20-50, 50-100 and 100-200 cm soil layers across the globe, respectively (Fig. 3). If Soil1-3 were included, additional 18%, 18%, 20%, and 13% of the variance could be explained in the four layers, respectively (Fig. 3). This result demonstrated that soil properties must be considered in order to explain the spatial variability of SOC stocks across the globe. However, it is noteworthy that the fitted model overestimated low SOC stocks, and underestimated high SOC stocks. This bias of model performance at the both ends of observed SOC stocks is common across all four depths (Fig. 3).

The results of the BRT model including soil properties (i.e., Soil1-3) indicated that Soil1 (i.e., the first PC of soil properties) was consistently the most important individual control of SOC stocks in the deeper three soil layers (i.e., 20-50, 50-100, and 100-200 cm; Fig. 4). On average, Soil1 alone contributed 21% (with 95% confidence intervals ranging from 17–24%), 23% (20–28%) and 22% (18–26%) to the explained variance of SOC stocks in the three deeper soil layers, respectively (Fig. 4). In the top 20 cm soil layer, Climate2 was the most important contributing 19% (15-23%) to the explained variance of SOC stocks; and Soil1 was the second most important and contributed 18% (16–20%). In the deeper three layers, the second most important contributors were NPP, biome type, and Climate3, respectively (Fig. 4).

Summing the relative importance of individual variables, the overall effect of soil properties was relatively consistent among the four layers, accounting for 30-40% of the overall influence of all assessed variables respectively, but were more important in the deepest two layers than in the top first layer (Fig. 5). The overall relative influence of climate was significantly higher than that of soil in the top 20 cm soil layer (43% vs 31%; Fig. 5). In the deeper three soil layers, the overall influences of climatic variables and soil properties were comparable and did not show significant difference. Overall, climatic variables accounted for 43% (38-47%), 36% (32-40%), 33% (28-37%) and 35% (31-39%) in the four layers, respectively; and soil properties accounted for 30% (27-33%), 35% (30-39%), 39% (35-43%), and 37% (33-41%), respectively (Fig. 5). The relevant influence of the left three variables (i.e., NPP, biome type and cultivation) was secondary and marginal (~10% in terms of relevant influence) compared to climate and soil variables, and together accounted for the remaining ~30% of the explained variance. With increasing soil depth, in general, the relevant influence of climate was decreased, while the influence of soil was increased. However, the overall influences of climate and soil remained relatively stable at the level of 70%. These results demonstrate the comparable and primary effects of climate and soil properties on SOC stocks.

**4 Discussion**

**4.1 The importance of soil properties**

A series of soil properties may directly or indirectly affect SOC dynamic processes via influencing carbon inputs to soil, microbial activity, and accessibility of carbon substrates to microbes, thereby SOC stocks. Sand content (which is the most important contributor to the first PC of soil), for example, has significant effects on the formation and transformation of soil aggregates which regulate the stability of SOC as well as soil porosity thereby oxygen availability for microbial decomposition of SOC (Dungait et al., 2012; Six et al., 2002). In addition, soil properties such as LL15 and DUL have dominant control over

soil water dynamics which further influence water availability for plant growth. Theoretically, LL15 is close to the minimal soil moisture required a plant does not wilt, it thus may strongly regulate plant growth therefore carbon inputs into soil and final SOC stocks. Together with DUL (i.e., drained upper limit – soil water content obtained at the matric potential of 33 kPa), LL15 determines the available water capacity of soil (AWC, i.e., the difference between DUL and LL15) and thus LL15 would affect SOC stock via its determination on soil AWC. While, AWC couples with a series of soil hydrological processes such as runoff and drainage, which have direct effects on the vertical/horizontal translocation of SOC (Luo et al.;Kaiser and Kalbitz, 2012). Soil properties are more important for controlling SOC stocks in deeper layers than in upper layers. This phenomenon may due to that soil structure may have substantial effects on water and oxygen diffusion in deeper layers. Potentially more frequent waterlogging and low oxygen in subsoil result in additional environmental constraints inhibiting microbial decomposition of SOC (Huang et al., 2020).

Our results demonstrate the primary control of soil properties on SOC stocks in the whole-soil profile across the globe. Indeed, the results suggested that soil-related principle components were consistently the most important individual influential variable in three deeper soil layers except in the assessed 0-20 cm soil layer. Soil physical and chemical properties directly determine the activity of decomposer community which mediates the decomposition of soil carbon (Derrien et al., 2014;Foesel et al., 2014;Bernard et al., 2012). More importantly, soil carbon can be physically protected from decomposition via occlusion with soil aggregates and binding with minerals (Lehmann and Kleber, 2015;Dungait et al., 2012;Schmidt et al., 2011), while the protection capacity is largely determined by soil physiochemical properties (Six *et al.* 2000). These physical protection processes may lead to soil-dependent stabilization/destabilization of different soil carbon substrates (Waldrop and Firestone, 2004;Keiluweit et al., 2015;Six et al., 2002). However, it should be noted that complex interplays of various soil properties are involved in SOC stabilization and destabilization processes. It is also difficult to obtain a cause-effect conclusion on the relationship between a particular soil physicochemical property and SOC stocks.

### 4.2 The importance of climate

Few studies have paid particular attention to the dynamics of SOC in subsoils across large scales. One might expect greater importance of climate in surface soils as topsoil is at the frontline of interacting with the atmosphere. But our results do not show a clearly decreasing importance of climate with soil depth. Rather, the overall influence of climatic variables on SOC stocks is statistically similar in all soil layers. In a forest soil, a recent study found that SOC in the whole soil profile down to 1 m is sensitive to warming (Pries et al., 2017). This sensitivity may be general across the globe. However, it is noteworthy that neither mean annual temperature nor mean annual precipitation were the most important individual climatic variables. Rather, climatic variables reflecting seasonal variability were more important. This result may suggest that, except average change trend, it is important to understand the change patterns of temperature and precipitation under climate change. For example, a number of studies have demonstrated that extreme climate events (e.g., drought and heatwaves) have significant effects on carbon cycle including soil carbon, due to their dramatic influence on the transport and availability of water and

energy as well as ecosystem functional processes (Reichstein et al., 2013). Field observations, particularly via manipulative experiments of whole soil profile, are certainly needed to detect how deep soil carbon responds to climate change as the result may have significant implications on the fate of deep soil carbon under future climatic conditions.

## 4.3 Secondary role of carbon inputs in determining spatial variability of SOC stocks

The effect of apparent carbon input, NPP, on SOC stock is generally small in all assessed soil layers (Fig. 5). This result is in line with findings from a continental-scale study across sub-Saharan Africa where climate and geochemistry are more important predictors of SOC content than aboveground carbon inputs (von Fromm et al., 2020). The importance of NPP may largely depend on how much NPP ends up in the soil and how it is translocated to different depths (Wang et al., 2021). Total NPP may not be a useful indicator of actual carbon inputs into different soil depths, particularly in deeper layers. Cultivation, for example, may substantially change the fate of plant biomass, a large fraction of plant biomass may be harvested as yield or consumed by livestock, and thus does not contribute to soil carbon. This could explain the phenomena that cultivation (cultivated vs non-cultivated in this study) and NPP show the similar importance in general. In addition, the final importance of carbon inputs may also depend on their quality (e.g., carbon to nitrogen ratio), while NPP alone does not bring such information. The quality of carbon inputs represented by their nutrient content and chemical structure plays a vital role in SOC formation and transformation (Hessen et al., 2004;Jastrow et al., 2007). In our study, biome type (which shows similar importance to NPP) would partially reflect the importance of carbon input quality as different biome types have distinct carbon biomass quality (e.g., wood vs leaf litters which are the main component of NPP). However, here we must to point out that the minor role of carbon inputs in determining the global spatial distribution of SOC stocks does not mean that they are not important for local carbon management. Under the same climatic and edaphic conditions, indeed, carbon inputs should be the predominant factor controlling if the soil is a carbon sink or source (Luo et al., 2017).

## 4.4 Limitations and future research

Although we have used a diverse and representative dataset across the globe for the analysis, there are still some limitations in the datasets and assessment. First, our study did not bring detailed land use history and intensity (such as the time length of cropping and the intensity of grazing) into the analysis, which may significantly affect SOC stabilization processes and thus SOC stocks in managed landscapes (Sanderman et al., 2017). As anthropogenic land use may change from year to year, it is challenging to accurately explain SOC stock changes in those systems that experience intensive human disturbances across large extents. Second, all soil properties including SOC were treated as constant. In reality, however, some soil properties, particularly chemical variables such as pH, may actively respond to external disturbance including human activities. Treating these variables as constant may result in under- or over-estimations of the variable importance if a variable shows marked temporal variability. Third, in managed systems, the apparent carbon input represented by NPP may not accurately reflect the real carbon input into soil (Luo et al., 2018;Pausch and Kuzyakov, 2018) as discussed above, leading to biased estimation of the importance of C inputs. In cropping areas, for example, yield harvesting and crop residue removal certainly reduce the

fraction of NPP ending up in the soil. Forth, we would like to point out that, albeit edaphic factors appear to be the dominant individual controls on SOC stock, climate might have an impact on those edaphic factors and hence SOC stocks in the long term (Jenny 1941). Indeed, Luo *et al.* (2017) have provided evidence that climate not only directly but also directly (via its effect on edaphic factors) exerts significant effect on SOC dynamics. All these limitations should be overcome to provide more robust predictions on the role of different factors in SOC stabilization and stock, which will be particularly important for understanding long-term SOC dynamics in managed systems. Fourth, we would like to note that this study focused on the controls over the global spatial pattern of SOC stocks and did not explicitly assessed the potential variability of controls at small scales. Under different land use types, for example, factors controlling SOC stock would change. A recent study focused on SOC component fractions have found that continental drivers of SOC stocks modulated by regional environmental factors (Viscarra Rossel et al., 2019). In order to better understand regional scale factors controlling SOC dynamics, we should further explore the controls over different spatial scales. Considering that we only included limited soil properties in our assessment and different soil properties may play different roles at different scales, scale-dependent understanding of controls over SOC stocks is important to make site-specific management practices for sustainable soil use and carbon management. Finally, soil samples are still limited in some areas (e.g., tundra and flooded grasslands and savannas) (Batjes, 2016). We do not know much about whether some of the relationships we find between SOC stocks and predictor variables are universal or maybe fundamentally different in less studied soils. The uneven distribution of soil samples may also help to explain the model bias in explaining low and high SOC stocks (Fig. 3). Our results indicate that there are large uncertainties in the relative importance of climate and soil depending on the data used to fit the model (Fig. 5).

## 5 Conclusions

Quantitatively, we have demonstrated the primary role of soil properties together with climate in regulating SOC stock in the whole soil profile across the globe. This result has important implications for understanding mechanisms of SOC stabilization and destabilization. Previous modelling and experimental efforts have mostly focused on climatic and biotic aspects, and many of the studies are over smaller scales. We argue that soil physicochemical characteristics define the boundary conditions for the climatic and biotic factors. That is, climatic and biotic factors (e.g., carbon inputs) can regulate the rate of SOC of shifting from one capacity to another, but a soil's physicochemical properties (e.g., soil structure) may inherently determine the SOC stock capacity of soil. It is thus critical to understand how soil processes mediated by different soil properties in different soil layers respond to those climatic and biotic factors and land management practices, and feed this information into the prediction of SOC stock capacity in the whole soil profile. However, individual soil variables work together involving complex interactions and non-linear relationships with each other as well as with climate to regulate SOC stock (Figs. 2 and 4). We need more and better quality data (e.g., following the same soil sampling and measuring procedure and using novel approach for monitoring of soil properties) and innovative methods (Viscarra Rossel et al., 2017) for representing soil heterogeneity to facilitate robust prediction of SOC dynamics over large extents. Results of this study further demonstrate that globally the influence of individual climatic variables on SOC stock is weaker than the influence of individual soil properties regardless of

soil depth. Current Earth system models are mostly driven by climate, with few cases have approximated the regulation of soil properties on carbon stabilization and destabilization (Tang and Riley, 2014;Riley et al., 2014). Undoubtedly, climate has direct effect on plant growth and thus potential carbon inputs to the soil, but our results demonstrate that soil properties are

also primary controls of global SOC stocks. Our research highlights the urgent need to consider soil properties and their interactions with climate to provide more reliable predictions of SOC stock and changes under climatic and land use changes.

## 6 Data availability

Soil data including soil organic carbon and the considered soil properties from WoSIS are available at http://www.isric.org/explore/wosis/accessing-wosis-derived-datasets. Climate data including 19 bioclimatic variables from

WorldClim are available at http://worldclim.org/version2. Code used to generate the results in this study can be reasonably requested from corresponding author.

## 7 Author Contribution

Z.L. conceived the study and assessed the data; Z.L. interpreted the results and wrote the manuscript with the contribution of R.A.V.R.; T.Q. contributed to data assessment and drawing figure.

## 8 Competing interests

The authors declare no competing interests.

## 9 Acknowledgments

We thank the people who originally collected the data and make this invaluable data publicly available. This study was financially supported by "the Fundamental Research Funds for the Central Universities" and "the Research Innovation

Foundation for Young Scholars" (K20200203).

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

415

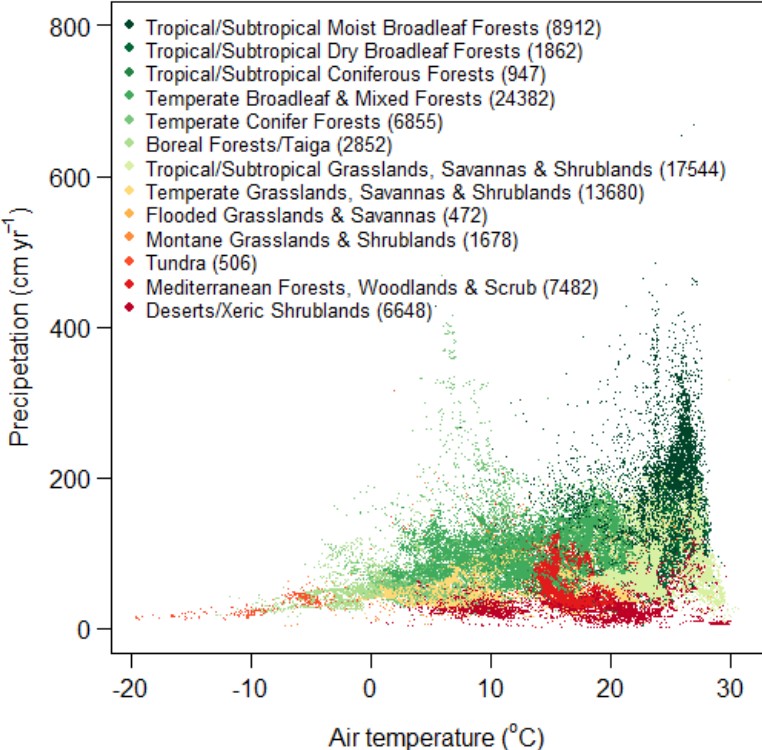

420

**Fig. 1. Distribution of soil profiles with soil carbon data in relation to mean annual air temperature and precipitation.**
Different colours show the biome type to which the soil profile belongs to. Numbers in parentheses show the number of soil profiles in the relevant biome. Some soil profiles (1382) were not included as climate and/or biome type could not be identified for them.

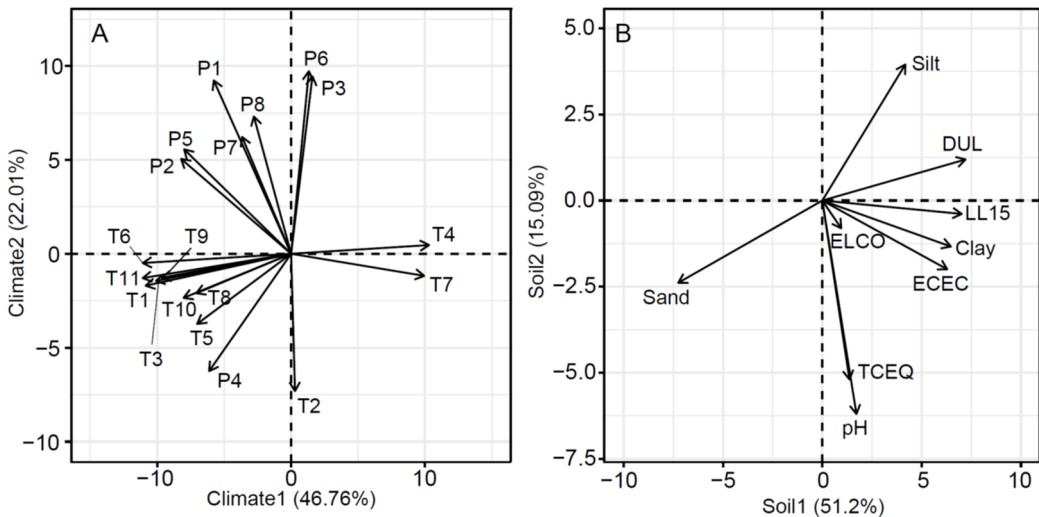

**Fig. 2. Loadings of 19 climatic variables (A)  and 9 soil properties (B) to the most important two principal components.**
T1, annual mean temperature; T2, mean diurnal range; T3, isothermality; T4, temperature seasonality; T5, max temperature of warmest month; T6, min temperature of coldest month; T7, temperature annual range; T8, mean temperature of wettest quarter; T9, mean temperature of driest quarter; T10, mean temperature of warmest quarter; T11, mean temperature of coldest quarter; P1, annual precipitation; P2, Precipitation of wettest month; P3, Precipitation of driest month; P4, precipitation seasonality; P5, precipitation of wettest quarter; P6, precipitation of driest quarter; P7, precipitation of warmest quarter; P8, Precipitation of coldest quarter. DUL, drained upper limit of soil; LL15, lower limit of soil; ELCO, electrical conductivity; ECEC, effective cation exchange capacity; TCEQ, calcium carbonate content; Sand, Silt and Clay, the fraction of sand, silt and clay content of soil; pH, soil pH. See Table 1 for more details about the variables.

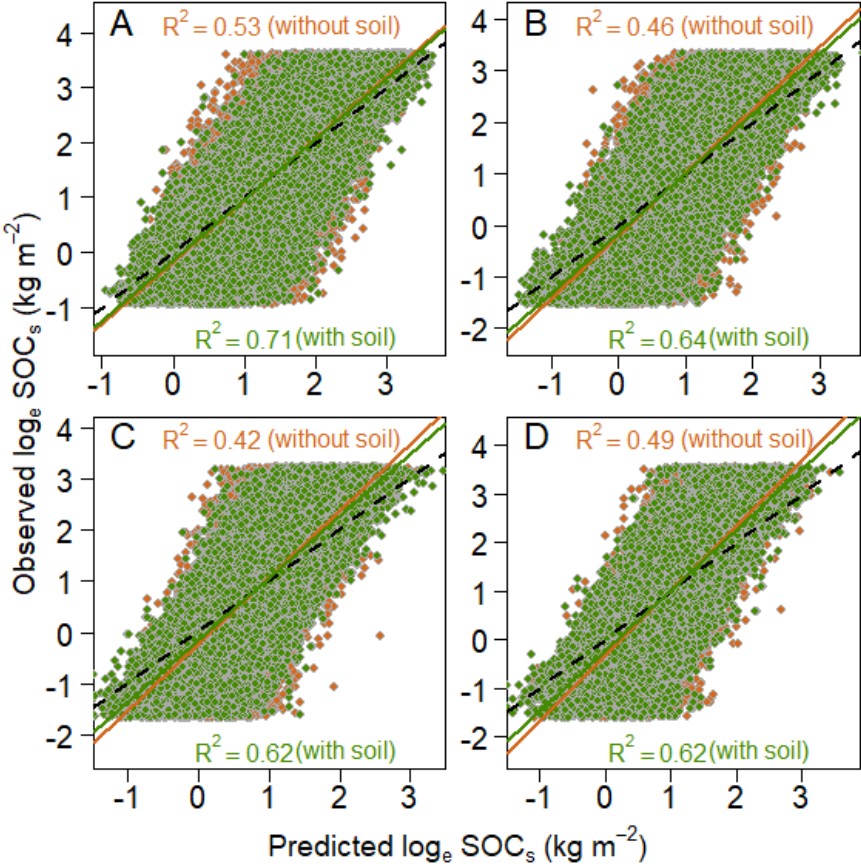

435

**Fig. 3. An example of the performance of boosted regression trees in explaining soil organic carbon stocks in four standard soil depths across the globe.** (A) 0–20 cm, (B) 20–50 cm, (C) 50–100 cm, and (D) 100–200 cm. The data was natural logarithm-transformed. The dashed line shows the 1:1 line. Chocolate and green circles show the results without and with predictors of soil properties, respectively.

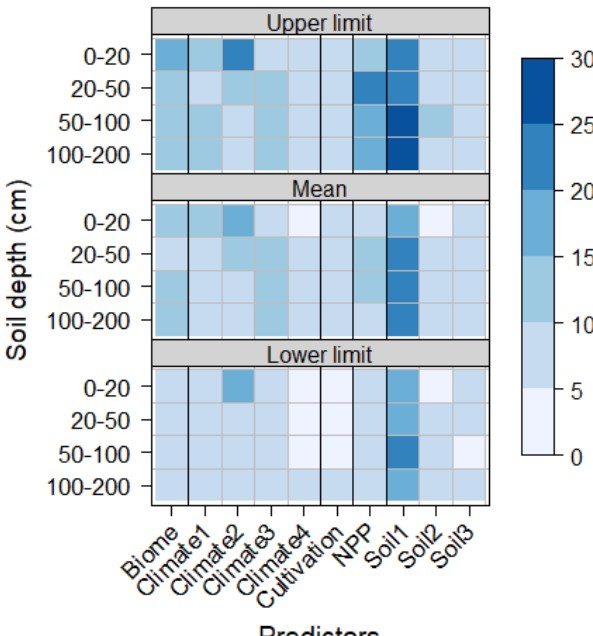

**Relative influence (%)**

**Fig.4. The relative influence of individual biotic, climatic and edaphic variables influencing global soil organic carbon stocks.** Upper limit (up panel), mean (middle panel) and lower limit (bottom panel) show the 97.5%, average, and 2.5% quantiles of 200 bootstrapping simulations, respectively. Biome, biome type; Climate1-4, the most important four principle components of 19 climatic variables; Cultivation, whether or not the land is cultivated (yes or no); NPP, net primary production; Soil1-3, the most important three principle components of 9 soil properties.

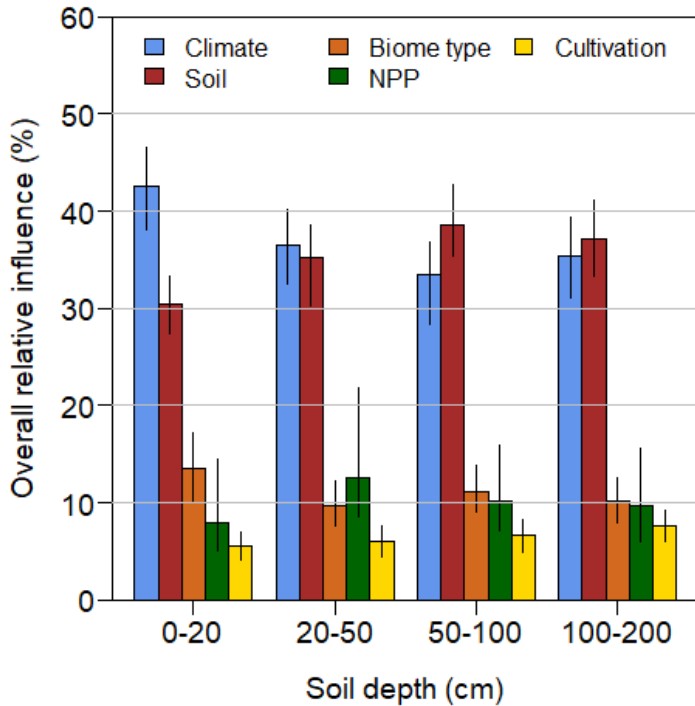

**Fig. 5. The overall relative influence of edaphic, climatic and biotic variables on soil organic carbon stocks in four soil depths across the globe.** The overall relative influence for climate and soil is calculated as the sum of the relative influence of their individual variables (which is shown in Fig. 4). Error bars show the 95% confidence interval estimated based 200 bootstrapping simulations.

450

**Table 1.** Covariates used in the modelling of soil carbon stocks across the globe.

| Covariates | Code | Description | Unit | Data sources |
|---|---|---|---|---|
| Soil properties | TCEQ | Calcium carbonate content | g kg$^{-1}$ | WoSIS (Batjes et al., 2017) |
| | ECEC | Effective cation exchange capacity | cmol kg$^{-1}$ | |
| | ELCO | Electrical conductivity | dS m$^{-1}$ | |
| | Clay | Clay content | % | |
| | Sand | Sand content | % | |
| | Silt | Silt content | % | |
| | pH | pH measured in $H_2O$ | - | |
| | LL15 | Lower limit obtained at a matric potential of 1,500 kPa | % | |
| | DUL | Drained upper limit obtained at a matric potential of 33 kPa | % | |
| Climatic variables | T1 | Annual mean temperature | °C | WorldClim (Fick and Hijmans, 2017) |
| | T2 | Mean diurnal range | °C | |
| | T3 | Isothermality (T2/T7×100) | % | |
| | T4 | Temperature seasonality (standard deviation×100) | °C | |
| | T5 | Max temperature of warmest month | °C | |
| | T6 | Min temperature of coldest month | °C | |
| | T7 | Temperature annual range (T5–T6) | °C | |
| | T8 | Mean temperature of wettest quarter | °C | |
| | T9 | Mean temperature of direst quarter | °C | |
| | T10 | Mean temperature of warmest quarter | °C | |
| | T11 | Mean temperature of coldest quarter | °C | |
| | P1 | Annual precipitation | mm | |
| | P2 | Precipitation of wettest month | mm | |
| | P3 | Precipitation of driest month | mm | |
| | P4 | Precipitation seasonality (coefficient of variation) | % | |
| | P5 | Precipitation of wettest quarter | mm | |
| | P6 | Precipitation of driest quarter | mm | |
| | P7 | Precipitation of warmest quarter | mm | |
| | P8 | Precipitation of coldest quarter | mm | |
| Other | Biome | Biome type | - | WWF (Olson et al., 2001) |
| | NPP | Net primary productivity | kg C m$^{-2}$ yr$^{-1}$ | MODIS (Zhao and Running, 2010) |
| | Cultivation | Whether the land is cultivated (yes or no) | - | MODIS (Channan et al., 2014) |