# Peer review of "Similar effects of edaphic and climatic factors on soil organic carbon stocks of the world"

_Biogeosciences, 2020_

## Referee Comment (RC1) · Anonymous Referee #1 · 16 Oct 2020

Review BG_2020_298 I have reviewed the Biogeosciences manuscript with the title " Soil properties override climate controls on global soil organic carbon stocks" by Luo & Viscarra-Rossel. The manuscript provides a data driven analyses on the controls of soil organic carbon stocks at the global scale using a data driven approach and a machine learning technique. The manuscript touches a timely issue, is well written and well structured. I also like how the authors have discussed their findings and constrained themselves from speculation, something that I find very important for correlation studies. Good job! My comments are mostly on clarification and some added context. Something that I would say requires a medium sized revision. Nothing dramatic, but probably requiring some additional analyses.

My main comments:

1. Subsidiary analyses: The author make a strong case for soil data to become more prominent at global scales for modeling soil C stocks in earth system models. However, I wonder how good the models actually work if you would leave out the soil data and let the other variables do the job. Probably also a quite strong model at the end. Have you checked for that?

Second question in that direction: You did PCA for the variables from worldclim but not for any edaphic variables. Why? They are also cross-correlated I would assume. Connected to this: I found the two very similar figures S4 and 2 almost bit confusing. Also because of the way you indicated you would use the findings between primary variables and PCA in l.179-181. I wonder if you might be better advised to bring in S4 into the main part and abandon Figure2. Similar comment for figures S6 and figure 3.

2. Uncertainty and global data distribution With a global dataset of that size you should be able to make some statements on the uncertainty of your assessment. For example, we all know that tropical soils or wetlands are still very underrepresented at the global scale. The map in the supplement cannot really tell us much about that issue in your study, but shows quite some empty space for boreal zones, for example. Can you give the reader some insight into how the dataset that you include is structured? What's the data distribution across climate zones and land use to name just two important factors? Is the depth distribution of observations for the most important target variables fairly reasonable for all those profiles?

Connected to this point, I think you need to revise figure 3 a bit. At least present the overall uncertainty behind these assessments of controls or (even better) give some idea on how and if this differs across certain areas of the globe.

3. Framing of the importance of identified controls Some framing on the identified controls and where across the globe they might be particularly important might be good. Some of them are universal, but for sure differ in strength across climate zones. Similarily, when discussing this dataset and going into some detail about what the output means I think you need to address that some controls are simply not included. For example, I was very surprised that you stress the importance of aggregation (which is very important of course) but you don't say much about pedogenic short range oxides, different clay minerals etc. These controls are very important and they also structure soils (and can build up aggregates). They differ greatly across the globe, too. So bringing soil into the global picture with the variables that you do is important, but you should stress that there is a long way to go. I highly recommend checking out the Ito & Wagai study from 2017 (Global distribution of clay-size minerals on land surface for biogeochemical and climatological studies. The maps he provides might be a very valuable addition to your assessment of potential controls and you could include them to make your case stronger.

Minor comments: - Some of the references cited in the text are not in the reference list. Please double check (Jenny 1994 for example). - L. 294 the second "directly" should be "indirectly" - Title states that the title that soil "overrides" climate. Maybe a bit too strong. I would say it has a more direct control on SOC than climate, but not necessarily overrides its. As the authors state themselves, that climatic influence can be direct and indirect, a statement that has also been propagated before by some of the cited references. - There are some minor grammar problems here and there. Should be fixed before sending the revision

---

## Referee Comment (RC2) · Anonymous Referee #2 · 26 Oct 2020

Soil properties override climate controls on global soil organic carbon stocks

This is a well-written and very worth-while study that will be of high interest to readers. There are a few grammatical issues that should be carefully checked before publication. I have some questions about the analyses that need clarification below.

(*Note, I was unable to open the supplemental materials file and it's possible that some of the information I'm asking for is there)

Biotic covariates- Is there any attempt to account for how different plant functional types contribute different amounts of their NPP to soil carbon, or is all NPP assumed to have the same contribution to soil C? Can this be accounted for by land cover type

somehow? A lot of NPP does not contribute much to SOC. For example, in DayCent the metabolic:structural ratio is used to estimate this, which is based on the lignin:N ratio of litter. The LiDEL model (Campbell et al., SBB) also provides another example of how litter chemistry can dictate the amount of soil C input from different types of plants

GBM model- It appears that the same edaphic factors were used to gap-fill missing BD values and SOC stocks (in the BRT model) as were used in the GBM model to determine the weight of influences of different factors on SOCs. Since the vast majority of the data was missing BD, doesn't this mean that the edaphic factors are over-weighted/double counted in your analysis?

PCA- the PCA of the climatic variables is a nice approach. Why didn't you do the same for the edaphic properties, since many of them are also co-variates?

Discussion- Is soil LL15 an edaphic property? Isn't it also related to climate and vegetation?

Does NPP have any greater influence on deep SOC in wetter environments than dry, indicating the importance of leaching in translocating plant inputs deeper into the soil? This would be very interesting to know.

Uncertainties and Limitations- Did you included agricultural and managed landscapes into one analysis? It seems like you should split converted/managed lands into a separate analysis from non-managed lands due to this large impact of disturbance that you discuss here.

---

## Author Comment (AC1) · 5 Nov 2020

I have reviewed the Biogeosciences manuscript with the title "Soil properties override climate controls on global soil organic carbon stocks" by Luo & Viscarra Rossel. The manuscript provides a data driven analyses on the controls of soil organic carbon stocks at the global scale using a data driven approach and a machine learning technique. The manuscript touches a timely issue, is well written and well structured. I also like how the authors have discussed their findings and constrained themselves from speculation, something that I find very important for correlation studies. Good job! My comments are mostly on clarification and some added context. Something that I would

say requires a medium sized revision. Nothing dramatic, but probably requiring some additional analyses.

Response: We appreciate these positive and encouraging comments. Here, we provide point-by-point response to each of the comments raised by the reviewer.

My main comments:

1. Subsidiary analyses: The author make a strong case for soil data to become more prominent at global scales for modeling soil C stocks in earth system models. However, I wonder how good the models actually work if you would leave out the soil data and let the other variables do the job. Probably also a quite strong model at the end. Have you checked for that? Second question in that direction: You did PCA for the variables from worldclim but not for any edaphic variables. Why? They are also cross-correlated I would assume. Connected to this: I found the two very similar figures S4 and 2 almost bit confusing. Also because of the way you indicated you would use the findings between primary variables and PCA in l.179-181. I wonder if you might be better advised to bring in S4 into the main part and abandon Figure2. Similar comment for figures S6 and figure 3.

Response: The suggestion on the check of leaving out the soil data and re-fitting the model is a good point. This re-assessment allows us to obtain direct evidence on the importance of climatic and edaphic variables as well as to confirm that whether the model was over-fitted. We will follow this suggestion to do some re-assessment in the formal revision.

The second question on the potential cross-correlation between edaphic variables is valid. We can conduct additional analysis to demonstrate the correlation between edaphic variables.

For other comments on the presentation of figures, they will be addressed.

2. Uncertainty and global data distribution With a global dataset of that size you should

be able to make some statements on the uncertainty of your assessment. For example, we all know that tropical soils or wetlands are still very underrepresented at the global scale. The map in the supplement cannot really tell us much about that issue in your study, but shows quite some empty space for boreal zones, for example. Can you give the reader some insight into how the dataset that you include is structured? What's the data distribution across climate zones and land use to name just two important factors? Is the depth distribution of observations for the most important target variables fairly reasonable for all those profiles? Connected to this point, I think you need to revise figure 3 a bit. At least present the overall uncertainty behind these assessments of controls or (even better) give some idea on how and if this differs across certain areas of the globe.

Response: Thanks for the suggestion to include discussion on the uncertainty induced by the complexity of the data. The reviewer provides good ideas on how to further explore the structure of the SOC dataset by climate zones, land use, depth and we will provide additional information in the revision.

3. Framing of the importance of identified controls Some framing on the identified controls and where across the globe they might be particularly important might be good. Some of them are universal, but for sure differ in strength across climate zones. Similarly, when discussing this dataset and going into some detail about what the output means I think you need to address that some controls are simply not included. For example, I was very surprised that you stress the importance of aggregation (which is very important of course) but you don't say much about pedogenic short range oxides, different clay minerals etc. These controls are very important and they also structure soils (and can build up aggregates). They differ greatly across the globe, too. So bringing soil into the global picture with the variables that you do is important, but you should stress that there is a long way to go. I highly recommend checking out the Ito & Wagai study from 2017 (Global distribution of clay-size minerals on land surface for biogeochemical and climatological studies. The maps he provides might be a very valuable

addition to your assessment of potential controls and you could include them to make your case stronger.

Response: We thank the reviewer for raising this point. We agree that other important variables are likely missing in our assessment. In the revision, we will expand the discussion on the importance of other missed, potentially important variables.

We checked the Ito & Wagai paper on the mapping of clay-size minerals across the globe. Their maps represent two layers: topsoil and subsoil. These are not consistent with the soil layers that we used in our study and the quality of the data would also not be consistent because our study uses measured data, not model estimates. For these reasons, it is very likely that we cannot explicitly include the information of clay composition in our potential revision. However, we will contact Ito & Wagai to request the relevant original datasets and check the consistency of their data with ours in terms of the measured soil locations.

Minor comments: - Some of the references cited in the text are not in the reference list. Please double check (Jenny 1994 for example). - L. 294 the second "directly" should be "indirectly" - Title states that the title that soil "overrides" climate. Maybe a bit too strong. I would say it has a more direct control on SOC than climate, but not necessarily overrides its. As the authors state themselves, that climatic influence can be direct and indirect, a statement that has also been propagated before by some of the cited references. - There are some minor grammar problems here and there. Should be fixed before sending the revision.

Response: Thanks for picking those up. We will check the manuscript carefully for the reference citations to ensure that the reference list and citation in the text are consistent. We will carefully re-check the language and statements made to ensure our expression is accurate and concise.

———————————————

---

## Author Comment (AC2) · 5 Nov 2020

This is a well-written and very worth-while study that will be of high interest to readers. There are a few grammatical issues that should be carefully checked before publication. I have some questions about the analyses that need clarification below.

Response: Thank you for these positive comments. We will carefully re-check for any grammatical issues. For the questions raised, below, we respond point-by-point.

(*Note, I was unable to open the supplemental materials file and it's possible that some of the information I'm asking for is there)

[Figure]

Biotic covariates- Is there any attempt to account for how different plant functional types contribute different amounts of their NPP to soil carbon, or is all NPP assumed to have the same contribution to soil C? Can this be accounted for by land cover type somehow? A lot of NPP does not contribute much to SOC. For example, in DayCent the metabolic:structural ratio is used to estimate this, which is based on the lignin:N ratio of litter. The LiDEL model (Campbell et al., SBB) also provides another example of how litter chemistry can dictate the amount of soil C input from different types of plants.

Response: These are good questions. We included land cover type as a predictor reflecting plant functional types. The land cover type data is from MODIS land cover product. The result is that land cover type is less important than total NPP. The reviewer is right that the contribution of NPP to soil carbon might be strongly dependent on plant functional types. The quality of NPP may be also important such as the nutrient content. Unfortunately, we do not have detailed data to test this. We will expand the discussion on the potential importance of plant traits.

GBM model- It appears that the same edaphic factors were used to gap-fill missing BD values and SOC stocks (in the BRT model) as were used in the GBM model to determine the weight of influences of different factors on SOCs. Since the vast majority of the data was missing BD, doesn't this mean that the edaphic factors are overweighted/double counted in your analysis?

Response: Thanks for raising this concern. We understand the reviewer's concern that the edaphic properties that had been used to infer BD had been double counted. Here, it should be noted that we only included several limited soil properties for SOC stocks (Table S1 in the manuscript), while 45 soil properties have been used due to the purpose of the GMB modelling is to fill missing BD. We will exclude those soil properties that will be used for SOC stocks to re-fit GBM models to predict BD. Considering that our analysis is at the global scale, this conduction would have little effect on the predictive power of the GBM model. This exercise will eliminate the double counting

issue.

PCA- the PCA of the climatic variables is a nice approach. Why didn't you do the same for the edaphic properties, since many of them are also co-variates?

Response: We saw that at least some of the climatic variables were highly correlated, while the the edaphic factors were less correlated. However, we agree that at least some of the edaphic properties will be correlated and so we will re-analyse the data to check and potentially also perform a PCA of these properties. Thank you for the comment.

Discussion- Is soil LL15 an edaphic property? Isn't it also related to climate and vegetation?

Response: Thanks for this question. In this study, LL15 is defined as lower limit under the pressure of 15 bar. It is inherently an edaphic property determined by soil texture and structure.

Does NPP have any greater influence on deep SOC in wetter environments than dry, indicating the importance of leaching in translocating plant inputs deeper into the soil? This would be very interesting to know.

Response: This is an interesting question. However, this study cannot explicitly quantify the importance of leaching in translocating carbon inputs. Also, please note that, in dry environments, roots may go deeper to find moisture. So, carbon transport due to leaching may be less in dry areas, but stimulated root growth would enhance root-derived carbon inputs in deeper layers. The effect of NPP may be complex, depending on plant functional types (Reviewer #1 mentioned this point), soil hydraulic properties, climate seasonality, etc. We will discuss the potential divergent effects of NPP on SOC stocks taking into account its interactions with plant functional type, soil and climate.

Uncertainties and Limitations- Did you included agricultural and managed landscapes into one analysis? It seems like you should split converted/managed lands into a separate analysis from non-managed lands due to this large impact of disturbance that you discuss here.

Response: Thanks for pointing out the importance of land management. Yes, we did not distinguish between natural and managed lands. We acknowledge that more detailed assessment is worth additional study. In the revision, we will highlight the importance of disturbance.
* * *

---

## Author Response (AR1)

**Response to reviewers' comments**

Reviewer #1

I have reviewed the Biogeosciences manuscript with the title "Soil properties override climate controls on global soil organic carbon stocks" by Luo & Viscarra Rossel. The manuscript provides a data driven analyses on the controls of soil organic carbon stocks at the global scale using a data driven approach and a machine learning technique. The manuscript touches a timely issue, is well written and well structured. I also like how the authors have discussed their findings and constrained themselves from speculation, something that I find very important for correlation studies. Good job! My comments are mostly on clarification and some added context. Something that I would say requires a medium sized revision. Nothing dramatic, but probably requiring some additional analyses.

Response: We appreciate these positive and encouraging comments. Following the relevant comments and suggestions, we have substantially revised the whole manuscript. Here, we provide point-by-point response to each of the comments raised by the reviewer.

My main comments:

1. Subsidiary analyses: The author make a strong case for soil data to become more prominent at global scales for modeling soil C stocks in earth system models. However, I wonder how good the models actually work if you would leave out the soil data and let the other variables do the job. Probably also a quite strong model at the end. Have you checked for that? Second question in that direction: You did PCA for the variables from worldclim but not for any edaphic variables. Why? They are also cross-correlated I would assume. Connected to this: I found the two very similar figures S4 and 2 almost bit confusing. Also because of the way you indicated you would use the findings between primary variables and PCA in l.179-181. I wonder if you might be better advised to bring in S4 into the main part and abandon Figure2. Similar comment for figures S6 and figure 3.

Response: The suggestion on the check of leaving out the soil data and re-fitting the model is a good point. This re-assessment allows us to obtain direct evidence on the importance of climatic and edaphic variables as well as to confirm that whether the model was over-fitted. We have followed this suggestion. The second question on the potential cross-correlation between edaphic variables is also valid. Particularly, the PCA for edaphic properties has been conducted. The models have re-performed by using PCA of both climatic and edaphic properties. In addition, following the reviewer's suggestion on excluding soil properties, we have fitted two sets of models with and without soil properties in order to directly demonstrate the importance of soil properties. The relevant results have been presented in Fig. 3. Based on the results, it is clear that, including climatic variables only, the model can explain less variance in global SOC stocks; while additional ~20% of the variance could be explained if soil properties have been included in the model (Fig. 3). Here, we also would like to note that using the principal component identified the PCA sacrifices model performance in terms of explaining SOC stocks. For example, $R^2$ in the top 20 cm soil reduced from 0.8 to 0.71.

For other comments on the presentation of figures, in this revision, we have thoroughly revised the data assessment by considering the reviewer's suggestions as responded above. The whole manuscript including **all figure presentations have been updated based on the new assessment**. Please refer to the revised manuscript for details.

2. Uncertainty and global data distribution. With a global dataset of that size you should be able to make some statements on the uncertainty of your assessment. For example, we all know that tropical soils or wetlands are still very underrepresented at the global scale. The map in the supplement cannot really tell us much about that issue in your study, but shows quite some empty space for boreal zones, for example. Can you give the reader some insight into how the dataset that you include is structured? What's the data distribution across climate zones and land use to name just two important factors? Is the depth distribution of observations for the most important target variables fairly reasonable for all those profiles? Connected to this point, I think you need to revise figure 3 a bit. At least present the overall uncertainty behind these assessments of controls or (even better) give some idea on how and if this differs across certain areas of the globe.

Response: Thanks for the suggestion to include discussion on the uncertainty induced by the complexity of the data. The reviewer provides good ideas on how to further explore the structure of the SOC dataset by climate zones, land use, depth. In this revision, we provided additional information in the revision. Specifically, we did the following two aspects to address these comments/suggestions.

First, we expanded the discussion on the limitation of the data distribution in terms of both geographic location and soil depth (lines 248-254). Particularly, we have provided a figure to show the distribution of data location in relation to mean annual temperature and precipitation (Fig. R1, Fig. 1 in the revised manuscript). It is clear that the data covers all biomes, albeit the data points in Tundra, flooded grasslands & savannas, and tropical/subtropical coniferous forests are less than 1000. We also provided some descriptive statistics on the data distribution among different biome types (lines 88-92).

[Figure]

Fig. R1. Distribution of soil profiles with soil carbon measurement in relation to mean annual air temperature and precipitation. Different colors show the biome type to which the soil profile belongs to. Numbers in parentheses show the number of soil profiles in the relevant biome.

Second, we presented the uncertainties in relative importance of individual variables (Fig. R2, Fig. 5 in the revised manuscript). We revised the approach to perform the BRT model. In order to quantify the uncertainties in the relative importance of soil, climate, etc. We conducted a bootstrapping simulation to obtain estimations of the 95% confidence interval of the relative importance of soil, climate, biome type, NPP and cultivation (lines 122-124, 139-140; Fig. 4 and 5). The whole result has been updated in the revised manuscript.

[Figure]

Fig. R2. The overall relative influence of edaphic, climatic and biotic variables on soil organic carbon stock in four soil depths across the globe. Error bars show the 95% confidence interval based on 200 bootstrapping simulations.

3. Framing of the importance of identified controls. Some framing on the identified controls and where across the globe they might be particularly important might be good. Some of them are universal, but for sure differ in strength across climate zones. Similarly, when discussing this dataset and going into some detail about what the output means I think you need to address that some controls are simply not included. For example, I was very surprised that you stress the importance of aggregation (which is very important of course) but you don't say much about pedogenic short range oxides, different clay minerals etc. These controls are very important and they also structure soils (and can build up aggregates). They differ greatly across the globe, too. So bringing soil into the global picture with the variables that you do is important, but you should stress that there is a long way to go. I highly recommend checking out the Ito & Wagai study from 2017 (Global distribution of clay-size minerals on land surface for biogeochemical and climatological studies. The maps he provides might be a very valuable addition to your assessment of potential controls and you could include them to make your case stronger.

Response: We thank the reviewer for raising this point. We agree that other important variables are likely missing in our assessment. In the revision, we expanded the discussion on the importance of other missed, potentially important variables (lines 253-255). In addition, we have included biome type, cultivation in the re-assessment. The relevant results were

shown in Figs. 4-5. In line with the reviewer's expectation, the importance of climate is universal, but biome types and cultivation also have significant effects, albeit their effects are secondary compared to the primary effect of soil and climate. For example, soil properties are more important in deeper layers than in upper layers. Based on the new results, we have updated the whole manuscript.

We checked the Ito & Wagai paper on the mapping of clay-size minerals across the globe. Their maps represent two layers: topsoil and subsoil. These are not consistent with the soil layers that we used in our study and the quality of the data would also not be consistent because our study uses measured data, not model estimates.

Minor comments: - Some of the references cited in the text are not in the reference list. Please double check (Jenny 1994 for example). - L. 294 the second "directly" should be "indirectly" - Title states that the title that soil "overrides" climate. Maybe a bit too strong. I would say it has a more direct control on SOC than climate, but not necessarily overrides its. As the authors state themselves, that climatic influence can be direct and indirect, a statement that has also been propagated before by some of the cited references. - There are some minor grammar problems here and there. Should be fixed before sending the revision.

Response: Thanks for picking those up. We have checked the manuscript carefully for the reference citations to ensure that the reference list and citation in the text are consistent. We also carefully re-checked the language and statements made to ensure our expression is accurate and concise. For the title, we have changed it to "*Comparable effects of soil properties and climate on soil organic carbon stocks across the globe*".

Reviewer #2

This is a well-written and very worth-while study that will be of high interest to readers. There are a few grammatical issues that should be carefully checked before publication. I have some questions about the analyses that need clarification below.

Response: Thank you for these positive comments. We have carefully re-check for any grammatical issues. For the questions raised, below, we respond point-by-point.

(*Note, I was unable to open the supplemental materials file and it's possible that some of the information I'm asking for is there)

Biotic covariates- Is there any attempt to account for how different plant functional types contribute different amounts of their NPP to soil carbon, or is all NPP assumed to have the same contribution to soil C? Can this be accounted for by land cover type somehow? A lot of NPP does not contribute much to SOC. For example, in DayCent the metabolic:structural ratio is used to estimate this, which is based on the lignin:N ratio of litter. The LiDEL model (Campbell et al., SBB) also provides another example of how litter chemistry can dictate the amount of soil C input from different types of plants.

Response:  These are good questions. We included land cover type as a predictor reflecting plant functional types. The land cover type data is from MODIS land cover product. The result is that land cover type is less important than total NPP. The reviewer is right that the contribution of NPP to soil carbon might be strongly dependent on plant functional types. The quality of NPP may be also important such as the nutrient content. Unfortunately, we do not have detailed data to test this. In this revision, we expanded the discussion on the potential importance of plant traits (lines 219-227).

GBM model- It appears that the same edaphic factors were used to gap-fill missing BD values and SOC stocks (in the BRT model) as were used in the GBM model to determine the weight of influences of different factors on SOCs. Since the vast majority of the data was missing BD, doesn't this mean that the edaphic factors are overweighted/ double counted in your analysis?

Response: Thanks for raising this concern. We understand the reviewer's concern that the edaphic properties that had been used to infer BD had been double counted (i.e., used for the BRT model for predicting SOC stocks). Here, it should be noted that we only included several limited soil properties for SOC stocks (Table 1 in the manuscript), while 45 soil properties have been used due to the purpose of the GMB modelling is to fill missing BD. In this revision, we have excluded those soil properties that will be used for SOC stocks to re-fit GBM models to predict BD. Considering that our analysis is at the global scale, this exclusion has little effect on the predictive power of the GBM model. This exercise eliminated the double counting issue. The whole data assessment has been updated based on this new idea.

PCA- the PCA of the climatic variables is a nice approach. Why didn't you do the same for the edaphic properties, since many of them are also co-variates?

Response: We saw that at least some of the climatic variables were highly correlated, while the edaphic factors were less correlated. However, we agree that at least some of the edaphic properties is correlated and so we re-analyzed the data to perform a PCA of these properties. See our detailed repose to a similar comment by Reviewer #1.

Discussion- Is soil LL15 an edaphic property? Isn't it also related to climate and vegetation?

Response: Thanks for this question. In this study, LL15 is defined as lower limit under the pressure of 15 bar. It is inherently an edaphic property determined by soil texture and structure, particularly in deep layers. However, we acknowledge that LL15 would be also correlated to climate, vegetation and even SOC itself, since any factors regulating soil formation and development may have to some extent direct or indirect effect on soil properties.

Does NPP have any greater influence on deep SOC in wetter environments than dry, indicating the importance of leaching in translocating plant inputs deeper into the soil? This would be very interesting to know.

Response: This is an interesting question. However, this study cannot explicitly quantify the importance of leaching in translocating carbon inputs. Also, please note that, in dry environments, roots may go deeper to find moisture. So, carbon transport due to leaching may be less in dry areas, but stimulated root growth would enhance root-derived carbon inputs in deeper layers. The effect of NPP may be complex, depending on plant functional types (Reviewer #1 mentioned this point), soil hydraulic properties, climate seasonality, etc. We discussed the potential divergent effects of NPP on SOC stocks taking into account its interactions with plant functional type, soil and climate (lines 219-227).

Uncertainties and Limitations- Did you included agricultural and managed landscapes into one analysis? It seems like you should split converted/managed lands into a separate analysis from non-managed lands due to this large impact of disturbance that you discuss here.

Response: Thanks for pointing out the importance of land management. Yes, we did not distinguish between natural and managed lands. We acknowledge that more detailed assessment is worth additional study. In the revision, we highlighted the importance of disturbance such as cultivation (see lines 219-222). Indeed, we have explicitly considered cultivation in our new assessment.

---

## Author Response (AR2)

Comments to the Author:

Dear Dr. Luo and co-authors,

Your manuscript has now been re-assessed by one of the reviewers, who thinks that you have largely addressed the previous comments, however also points out that the results section should be elaborated in some more detail. Furthermore, the discussion would profit from some streamlining and the supplement requires some revision and needs to be clearly referenced in the main manuscript. The reviewer made also some further excellent suggestions, which I ask you to address in your revision. Furthermore, please make sure that also Fig. 4 is self-explaining by specifying the variables directly in the Figure caption, instead of referring to Table 1.

I look forward to seeing your thoroughly revised manuscript.

Best regards,

Michael Bahn

**Response:**

Dear Dr. Bahn,

Thank you very much for considering our manuscript and your constructive comments and suggestions. Following the suggestions/comments by you and the reviewer, we further revised the manuscript. Briefly, we made the following three aspects of improvements by considering the comments on the results, discussion and self-explaining of figures (both Fig. 2 and Fig. 4). In the result section, some additional details have been included. The discussion section has been also revised. For details of our responses to the relevant comments, please refer to our point-by-point responses below.

We believe our revision has addressed all the comments from you and the reviewer. Look forward to hearing from you at your earliest convenience.

Best regards,

Zhongkui Luo (on behalf of all co-authors)

Review Luo et al. Biogeosciences (Revision 1)

I have reviewed the revised version of Luo et al. with the title "Similar effects of edaphic and climatic factors on soil organic carbon stocks of the world" submitted to Biogeosciences. I was also reviewer for the original submission and I have reviewed both the response letter as well as the revised submitted manuscript files. In general the authors did a good job in revising the manuscript. I don't have many remaining comments. Most of them are of minor size and should be easy to address. I find the discussion in parts a bit too long and speculative but this can be easily addressed. What I was confused about was the large amount of supplementary material that is not referred to in the text, even though it might be important. Please find my comments below. Line references refer to the revised manuscript file.

**Response:** We appreciate the positive comments on our first revision. The manuscript has been further revised by considering the comments by the reviewer. Please refer to the point-by-point response below.

Results section:

1. The results section is very short. And I think you need to explain the loading of the different PCs in a better way. You could also create a table for that shows the correlation of independent predictors with each PC, cumulative variance explained, Eigenvalue of PC etc. To me it seems a bit unclear from the fig 2 and description what really the most important soil variables were and how they cross-correlated with other variables.

**Response:** We understand the reviewer's point. However, we think the information required by the reviewer have been shown in the figure (i.e., Fig. 2). According to the nature of PCA, the length and direction of the arrows for each individual variable show the loadings to the principle components. For example, the loadings of T7 to Climate1 and Climate2 are 10 and 1.2, respectively (Fig. 2A). While, the degree of the angle between two arrows represents the correlations (i.e., 0° indicates a positive correlation with a correlation coefficient of 1, 180° indicates a negative correlation with correlation coefficient of -1).

For example, P6 and P3 are closely positively correlated. For these reasons, we think that Fig. 2 is appropriate and includes the necessary information.

2. The uncertainty section regarding model testing etc. mentioned in the response letter might be well suited to be a separate results section.

**Response:** The uncertainty has been described with the relevant averages (lines 169-172 in the clean version of the revised manuscript). In addition, we have discussed the relevant uncertainties in section 4.4 (lines 265-266).

3. Also a small section on the model performance in different biomes or climate zones might be interesting in this regard. (Fig 3 shows some weird bias in performance, see comment there.

**Response:** Thanks for this suggestion. In this study, we did not fit typical models for each biome or climate zone. Rather, we treated biome type and climate as determinants of SOC stocks. So this study cannot provide biome- or climate zone-specific model performance. However, we have discussed the importance of scale-dependent understanding of underlying drivers in lines 254-261.

Discussion                                                                                                                                          section:
1. Discussion: 4.1. this section is a bit speculative. You describe in the results section that predominantly texture variables and pH were the most important explanatory factors. Yet you begin the discussion with mediation of decomposer communities by these variables, stabilization mechanisms etc. I would think you better start this section with what you wrote in l. 187 and so on before you go into all the stabilization mechanism and microbio stuff and be very careful to state that your variables actually mean indicate that. It is good that you try to interpret what mechanisms are behind the variables that the model selects but you should add a sentence here that interpretation has to be done with great caution or that we need to address/measure certain aspects of the linkages of these soil variables to mechanism carefully (with examples of changing relationships of soil variables across climate or land use zones from the literature ). To stay with the example of pH and texture: this could mean all of what you write here is true, or maybe nothing as these variables integrate so many effects at the same time.

**Response:** We appreciate this suggestion on the structure of section 4.1 and the interpretation of mechanisms. Following the suggestion, we reorganized section 4.1. To emphasize the importance of the linkages of soil variables, we added the following statement at the end of this section: "*However, it should be noted that complex interplays of various soil properties are involved in SOC stabilization and destabilization processes. It is also difficult to obtain a cause-effect conclusion on the relationship between a particular soil physicochemical property and SOC stocks.*"

2. Discussion section 4.3 (l. 218 .227). This is super interesting. I read something similar in a discussion paper by Fromm et al. currently reviewed in SOIL. You could check that paper and also the references cited in there (which you should refer to regarding that this section of the discussion is not well underpinned with literature yet). I do share your interpretation that most soil systems, on the long term show SOC stocks not to be limited by annual C input but by how well C can be stabilized in soil. Make sure this is understood and maybe add something on that in this discussion section 4.3.

**Response:** Thanks for the recommendation to include reference citations to support our interpretation. Indeed, we were excited about the finding that the importance of NPP is limited, which promotes us to specifically quantify the direct contribution of annual carbon input to bulk soil carbon pool in another study. The results are very interesting. Please find the preprint of our new study here in case you are interested in: https://www.researchsquare.com/article/rs-65178/v1. In this study, we further expanded the discussion by considering the suggestions by the reviewer. Particularly, more references are included to support our statements. Following the suggestions by the reviewer, the paper by Fromm et al. (2021) and some reference have been cited.

3. Discussion section 4.4 (l.233-249). I would add to this section that for many areas we simple lag behind with gathering soil data. Just look at the bias that your dataset has towards the temperate zone. Many soil types are simple not well constrained

and we don't know much about whether some of the relationships we find between C stocks and predictor variables are universal or maybe fundamentally different in less studied soils (dryland, tropical soils, melting permafrost soils, developing or degraded soils, to name just a few examples).

**Response:** This is a good point for discussion. We added this discussion in section 4.4 (see lines 261-264).

Figures:

1. Figure 2. I think you need to add a legend that explains the abbreviations directly to the figure and not refer to table 1. It is simply too many variables.

**Response:** Thanks for this suggestion. Added accordingly.

2. Figure 3. Fig 3 shows some weird underperformance bias for high observed log SOC ,consistent across all four depths. Explanation? Is this discussed? Maybe some part of the data consists of soil data points that are not covered by the environmental variables used here ? for example peat or wetland soils that are not well predicted (= constrain to model performance)?

**Response:** Thanks for this insightful suggestion. We have mentioned this point in the result (lines 156-158) and discussed the potential underlying reasons (lines 261-265).

Supplement:

I think the authors must have missed to revise parts of the supplement. Fig 3-5 in the main manuscript appear in a similar form in the supplement, similar for one of the tables. In addition, none of the figures in the supplement are referred to anywhere in the text. There is also additional analyses shown (partial dependence plots) that are not reflected on in the manuscript, even though they are important and interesting for the reader. Can you explain and address that?

**Response:** We are sorry for the confusion induced by the supplemental materials. In the revision, indeed there was no supplement. In the submission system, we would have forgotten to remove the original supplementary file. In our original submission, there was a partial dependence plot. In the revision, however, we have deleted that plot due to that we focused on the PCA rather than the individual soil or climate variables by considering the comments by the two reviewers.

Minor comments

L 42. Space missing in the reference list.
l. 55. I would replace "cultivation" with land cover or land use.
l. 84-85. What you write within [] I would put as a separate sentence following the one you have it currently included in.
l. 101 delete "the" at: at the strict steady state
l. 105 delete "also"
l. 138-139. Remove the two "is".

**Response:** Thanks for these editing suggestions. Changed accordingly.

l.143: "land use" instead of "cultivation" ?

**Response:** Cultivation is a type of land use. Here, the correct word is "cultivation".
l. 165: "," after "respectively.
l. 173. Revise wording. Make two sentences, one for the decrease if climate, one for the increase of soil influence
l. 181. Spaces missing between references. I stopped checking here as the problem seems to be also in many other places).

l. 198 grammar.

**Response:** Thank the reviewer for her/his careful review. Changed or revised accordingly.

[revised manuscript text omitted]